# Surface Magnetostriction of FeCoB Amorphous Ribbons Analyzed Using Magneto-Optical Kerr Microscopy

**DOI:** 10.3390/ma13020257

**Published:** 2020-01-07

**Authors:** Kamila Hrabovská, Ondřej Životský, Jaroslav Rojíček, Martin Fusek, Vratislav Mareš, Yvonna Jirásková

**Affiliations:** 1Department of Physics, Faculty of Electrical Engineering and Computer Science, VŠB—Technical University of Ostrava, 17. listopadu 2172/15, 708 00 Ostrava-Poruba, Czech Republic; ondrej.zivotsky@vsb.cz; 2Research Center of Liquid Metal Physics, Boris Yeltsin Ural Federal University, Ekaterinburg 620002, Russia; 3Department of Applied Mechanics, Faculty of Mechanical Engineering, VŠB—Technical University of Ostrava, 17. listopadu 2172/15, 708 00 Ostrava-Poruba, Czech Republic; jaroslav.rojicek@vsb.cz (J.R.); martin.fusek@vsb.cz (M.F.); 4Center for Advanced Innovation Technologies, VŠB—Technical University of Ostrava, 17. listopadu 2172/15, 708 00 Ostrava-Poruba, Czech Republic; vratislav.mares@vsb.cz; 5CEITEC IPM, Institute of Physics of Materials, AS CR, Zizkova 22, 616 00 Brno, Czech Republic; jirasko@ipm.cz

**Keywords:** magneto-optical Kerr microscopy, domain imaging, magneto-elastic effect, finite element method, surface magneostrictive coefficient

## Abstract

Surface sensitive magneto-optical Kerr microscopy completed with the special self-made sample holder is used for studying the magneto-elastic behaviour in the surface of the as-quenched amorphous Fe_73_Co_12_B_15_ alloy. The 10, 5, and 3 mm wide and approximately 34 μm thick ribbons were prepared by the conventional planar flow casting process. The experimental setup allows for a simultaneous application of an external magnetic field in the directions parallel and perpendicular to the ribbon axis and of compression stress from one side of the sample, resulting in tensile stress in opposite side. The distributions of tensile stresses in the measured surface were modelled by the finite element method. The observed changes of the magnetic domains and hysteresis loop anisotropy field under applied stress are evaluated using the Becker–Kersten method. This resulted in the determination of the local surface magnetostrictive coefficient from an area of about 200 μm in diameter. The obtained values ranged between 37–60 ppm and were well comparable with the bulk value presented in the literature.

## 1. Introduction

Magnetostriction is a physical property of magnetic materials changing their shape in dependence on an applied magnetic field. It is a ubiquitous phenomenon in solid materials reaching a wide range of values, between 10^−8^ (strongly correlated systems) up to 10^−2^ (rare earth and other intermetallics). Applications in sensors and actuators, among many others, have been known since the 1970s [1]. Magnetostriction is expressed for any material by the magnetostrictive coefficient *λ* depending on the intensity of the applied magnetic field. If the strength of the magnetic field is high enough to magnetically saturate the material, the *λ* coefficient reaches its maximal value denoted as the saturation magnetostrictive coefficient *λ*_s_ [2]. It is constant and frequently used as a material characteristic.

Generally, two kinds of magnetostriction are known—directional or linear and volume [3]. Directional magnetostriction occurs in materials elongating (positive *λ*_s_) or shrinking (negative *λ*_s_) in the direction of the applied magnetic field, whereas the volume of material remains nearly unchanged. Contrary to this, volume magnetostriction is related to the whole material. In crystalline or polycrystalline materials, the magnetostriction values depend on the crystallographic direction of measurement and on the direction along which the magnetization is oriented by the applied magnetic field. The magnetostriction measured along the applied field, called parallel, is different from that, termed perpendicular, measured along the perpendicular direction. Both determine the volume magnetostriction. The difference between the parallel and perpendicular magnetostriction determines the so-called shape magnetostriction.

The inverse phenomenon to the mentioned directional and volume magnetostriction is a magneto-elastic effect. Mechanical stresses, e.g., tensile stress, planar compressive stress, etc., influence the values and signs of the magnetostrictive coefficients which are reflected in changes of magnetization [3]. This effect is often applied in connection with alternate current excitations. Induced mechanical oscillations are then used for high-performance ultrasonic generators or for electric-mechanic components working as band-pass filters [4]. Currently, there is fast development in the market of high-magnetostrictive films [5], especially multilayer thin foils that are employed in the controllers of microsystem technology as a sensor material. The requirements for these optimized soft magnetic films are high magnetostriction and low values of the coercive field and saturation magnetization. However, the size and sign of λ are important parameters reflecting the domain structure, the parameters of the stress-dependent hysteresis loop, and the magnetization processes in the material.

The shape and volume magnetostrictive coefficient can be measured by direct and indirect methods [3]. Direct methods include, for example, experiments with tension meters, capacity sensors, or interferometers by means of optical components. The main disadvantage of these techniques is their impossibility to establish the magnetostrictive coefficient in saturation. One of the indirect methods is the Becker–Kersten (BK) method [6], which is based on the measurements of the hysteresis loop in dependence on the applied external stress and provides accurate results for the volume coefficient of magnetostriction, especially for the magnetically soft amorphous and nanocrystalline ribbons prepared using planar flow casting [7,8]. This technique could be also used for new Fe-based metallic glasses prepared by 3D print technology [9].

Present studies are devoted to applying the Becker–Kersten method in the evaluation of magnetic characteristics measured in the close surface layers of the amorphous Fe_73_Co_12_B_15_ amorphous ribbon using magneto-optical Kerr microscopy. The first possibilities to find local surface *λ*_s_ using the magneto-optical Kerr effect were published in [10]. Because the surface magnetic properties of amorphous ribbons are often locally inhomogeneous, the visualization of the place on the surface into which the laser spot was focused was difficult. This weak point of the mentioned paper is here overcome by applying magneto-optical Kerr microscopy. It has many advantages, such as being contact-less and enabling the direct visualization of the illuminated area, the observation of magnetic domains, and the depiction of the surface hysteresis loop under applied stress. To simultaneously observe the changes in the magnetic domain structure and hysteresis loops under applied stress, the special sample holder was constructed and located within the Kerr microscope as a common experimental setup. This has enabled to analyse both the positive and negative surface saturation magnetostriction values of the ribbon-type samples.

## 2. Experimental Details

### 2.1. Experimental Material

The as-prepared Fe-Co-B amorphous sample of nominal composition 73 at.% Fe–12 at.% Co–15 at.% B was prepared from the pure input materials using planar flow casting (PFC) technology in air into a form of a 34 μm thick and 10 mm wide ribbon. The surface roughness, defined by two main parameters *R*a (average arithmetic deviation of the profile) and *R*z (maximal height of the profile), was measured using the Mitutoyo contact profile, SurfTest SJ-301 (Mitutoyo corporation, Kanagawa, Japan) and using a confocal microscope Olympus LEXT 3100 (Olympus Corporation, Tokyo, Japan), see Figure 1. The parameters *R*a ~ 0.398 μm and *R*z ~ 2.710 μm (Mitutoyo) and *R*a ~ 0.301 μm, *R*z ~ 4.467 μm (Olympus) were obtained for the ribbon surface being during PFC production in contact with the rotating wheel and denoted as the “wheel” side. The opposite, so called “air” side, was markedly smoother, which was reflected by both methods; *R*a ~ 0.186 μm and *R*z ~ 0.886 μm and *R*a = 0.155 μm, *R*z = 0.904 μm, respectively. The quality of the air surface was satisfactory for the Kerr microscope (evico magnetics GmbH, Dresden, Germany) observations. The studies were done on the samples of dimensions *R*_A_ (length) × *R*_B_ (width) = 8 × 10, 8 × 5, and 8 × 3 mm^2^ (see Figure 4).

### 2.2. Experimental Methods

The amorphous structures of the ribbon-type samples were checked by the X-ray diffractometer EMPYREAN (Malvern Panalytical Ltd., Malvern, UK) with Co Kα radiation (*λ* = 0.17902 nm) in the 2θ range from 20° to 140° in steps of 0.008° and time per step 500 s. A FEI Quanta 650 FEG scanning electron microscope (Thermo Fisher Scientific, Hilsboro, OR, USA) working at accelerating voltage of 10 kV and equipped with a detector for energy dispersive X-ray analysis (EDX), was used to follow the chemical composition.

The basic bulk magnetic characteristics were measured using the vibrating sample magnetometer (VSM, MicroSense, MA, USA) Microsense EZ 9. The local surface magnetic characteristics, namely the magnetic domains in static magnetic fields and hysteresis curves, were measured by the experimental setup which is schematically depicted in Figure 2. It consisted of the Kerr microscope based on the polarization Zeiss microscope AxioImager M1 completed with the newly designed sample holder seen in Figure 3 which together with accessories allowed for the controlling of the sample deflection and thereby induced the tensile stress in the air surface of the ribbon sample. White light from a Xe lamp was used as a light source.

## 3. Results and Discussion

### 3.1. Experimental Setup and Sample Holder

In order to find the local surface saturation coefficient of magnetostriction, the sample holder described below was implemented into the magneto-optical Kerr microscopy equipment (Figure 2). This surface-sensitive non-destructive method enables the observation of magnetic domains in the illuminated local surface area and simultaneously the imaging of the corresponding hysteresis cycle [11,12]. Moreover, the position of the light incident plane can be adjusted using the aperture diaphragm in the back focal plane of microscope. In such a way, the sensitivity can be set to in-plane longitudinal and transversal magnetization components or to out-of-plane polar magnetization components without the necessity to rotate with the sample or with the magnetic field [13]. Due to this fact, the Becker–Kersten technique [6] can be easily used for the material with positive magnetostriction presented here.

Because the Kerr contrast is a relatively weak effect, it requires an image contrast enhancement. Firstly, the measured sample was saturated in sufficient magnetic field *H* and its image of the surface (reference) was captured and stored. Subsequently, the magnitude of the magnetic field was decreased and the actual surface image was in real time subtracted from the reference, showing magnetic domain patterns optimally without the topographical contrast. Electronic noise was reduced by the number of averages. The standard magneto-optical hysteresis loop from the illuminated area was obtained by plotting the averaged Kerr intensity as a function of the magnetic field applied up to ±80 kA/m with the step 80 A/m. This image was normalized to the maximum of the absolute intensity value, giving the range from −1 to 1. The magnetic domain patterns were received by the KerrLab software at each point of the loop. The sample area from which the surface hysteresis loops were measured is a circle of 200 μm in diameter.

The brass non-magnetic sample holder in Figure 3 was positioned together with the central stand between a pair of rotating magnets. The sample was placed on the upper side of the holder, which had two grooves at the sides. The sample edges were fixed from above by a retaining washer with a hole and protrusions extending into the holder grooves. The sample and retaining washer were fixed against movement by a lock nut. This arrangement, schematically shown in Figure 3 right, enabled the white light of the Xe lamp of the microscope (Figure 2) to impact the air-side of the sample and consequently be reflected. During the experiment, the wheel-side of the sample came into contact with the micrometric screw. Its movement up caused the sample deflection and produced tensile stress in the top layer of the sample air-side. The compressive force *F* of the sliding contact acting on the wheel ribbon side was monitored by a strain gauge located in the middle of the holder and displayed by the Newport digital monitor.

Relation *σ* = *F*/*S* cannot be used for the calculation of the tensile stress in the sample surface layer, because in reality the entire sample is exposed to the bending stress. Therefore, the distribution of the biaxial tensile stress in the measured surface was modelled by the finite element method (FEM). This is a well-known approximate numerical method used for the solution of problems, mainly from the area of solid mechanics. In place of a continuous model of solids, a discrete model was applied and the continuum was replaced by the nodes and final elements [14]. The simulations were done by using the commercial software Marc with the MSC software Mentat pre-processor working on the Windows 7 operating system. The computing hardware contains a standard personal computer with the following parameters: Intel^®^ Core CTM i7 3770 CPU 3.40 GHz processor, 32.0 GB operating system memory.

Because the investigated ribbon-type samples were thin (~34 μm), the FEM was based on the shell model [15]. Its advantages include its lower size and fast calculation speed but, on the other hand, the more complex evaluation of the results due to the simplification of the structure geometry is the main disadvantage. The thickness of the sample was divided into five layers of equal thicknesses of 6.8 μm. The penetration depth of a light in metals (about tens of nm) determines the magneto-optical Kerr effect (MOKE) loop and domain measurements, reflecting therefore the magnetic properties of the top surface region. To determine the local surface saturation coefficient of magnetostriction, the values of tensile stress in the top fifth surface layer were taken into account. The stress–strain state of the sample was solved using four-node elements (quad 4) labelled as No. 75 in the library of elements in the Marc program [16]. Motivated by the experimental data, the material of the examined FeCoB ribbon was simulated by a linear model. With regard to the sample dimension, measured displacements, and contacts used, two types of non-linearity, namely “large displacement” and “structural non-linearity” were considered. This task was solved as dynamic with inertial forces incorporated. The numerical solution of the system of non-linear equations was done using the Newton–Raphson method [16,17].

The retaining washer and the contact surface of the micrometric screw were modelled as absolutely rigid surfaces, while the ribbon-type sample as a deformable shell type 75 [16]. Between the bodies the frictionless contact is expected. The ribbon-type sample was subjected to small strain due to its fixing by the retaining washer and lock nut. Nevertheless, this force, presented in Table 1, is low. The main sample deformation is due to a movement of micrometric screw with a linearly increasing pressure force towards the wheel-side of the sample to maximum load (Table 2).

Figure 4 depicts the dimensions of the ribbon-type sample; length *R*_A_, width *R*_B_, and thickness *R*_T_ of the micrometric screw, *S*_A_, *S*_B_, *S*_R_, and of the retaining washer, *W*_A_, *W*_B_. The concrete values used to model the tensile stress on the sample surface are summarized in Table 1. It also contains other details of the applied numerical model, including the pressure force of the washer, the size of the elements, the number of elements, and the number of nodes in the network. The material model is assumed to be linear, homogeneous, and isotropic with Poisson’s number *μ* = 0.3. Young’s modulus *E* = 150 GPa was measured by the ZWICK/ROELL Z150 tensile testing machine at room temperature.

Figure 5 illustrates an inhomogeneous distribution of the tensile stresses in the modelled top fifth surface layer of the FeCoB samples. It shows only ¼ of the task (ribbon); the centre of the ribbon is situated in the origin of the coordinate system. The left subplot depicts the sample (10 × 8) mm^2^ in the case when the pressure force of micrometric screw is adjusted to 3.92 N. Because the sample is fixed from all four sides, the biaxial tensile stress is observed. The middle and right subplots correspond to the samples with dimensions (5 × 8) and (3 × 8) mm^2^. Here, the distribution of stresses is much more homogeneous, due to the fixation of the samples only from the opposite two sides. Because here we are close to applied uniaxial stress, only the *σ*_x_ dependences are presented. Figure 6 shows the modelled tensile stresses in the *x* and *y* axes for the same samples and forces of the micrometric screw as in Figure 5 from the area of one element (side size 100 μm) on the surface of the fifth layer. It is evident that the stresses in the black rectangular bounded area are homogeneous. The calculated values of *σ*_x_ and *σ*_y_ for different screw forces are summarized in Table 2. Whereas for the sample (10 × 8) mm^2^, *σ*_x_ < *σ*_y_, in samples (5 × 8) mm^2^ and (3 × 8) mm^2^ the *σ*_x_ exceeds *σ*_y_ approximately 3–4 times and 4–9 times, respectively. Thus it follows that the stronger uniaxial stress at the narrower samples is induced in direction of the *x*-axis.

### 3.2. Sample Characterization

The amorphous state of both sides of the Fe_73_Co_12_B_15_ ribbon was confirmed by X-ray diffraction (XRD). Both diffractograms were identical and therefore only the pattern from the air side is shown in Figure 7a. It consists of only broad amorphous halos at the angles of 2θ = 52.41° and 94.20°. The chemical composition as determined by EDX analysis—Fe 68.5 at.%–Co 11.9 at.%–B 19.6 at.%—reflects a small increase of B at the expense of Fe (Figure 7b) in comparison with the nominal one.

The bulk magnetic parameters of the FeCoB material were obtained from the hysteresis loops measured by VSM. They are completed by bulk saturation magnetostriction coefficient, *λ*_sb_, obtained from [18] and summarized in Table 3.

Figure 8 represents the surface magnetic properties of FeCoB sample obtained using the magneto-optical Kerr microscopy without the applied stress. The sample with dimensions (10 × 8) mm^2^ was fixed on a standard sample holder and its magnetic domains and magnetization curve were measured in longitudinal MOKE configuration. This means that the magnetic field, the plane of light incidence, and the ribbon axis are mutually parallel. The typical rectangular hysteresis loop and wide strip domain with the wall almost parallel to the applied magnetic field indicate that the local easy magnetization axis lies along the ribbon axis. Subplots (a), (b), and (c) show consecutive images of the domain patterns in increasing positive magnetic fields, corresponding to the reversal of the measured magnetization curve. The surface coercive field ~1.76 kA/m confirms the magnetic softness of the sample; nevertheless, it is approximately two orders higher as compared to the bulk value, 15.12 A/m. This difference can be ascribed to surface inhomogeneities, very thin oxide layers, and/or microcrystals that can be dispersed in small amounts only in the close surface layers which are not visible by XRD. A visible hysteresis loop noise is caused by the roughness of the ribbon surface.

### 3.3. Tensile Stress and Saturation Magnetostrictive Coefficient

It is known that stress applied to magnetostrictive materials induces uniaxial magnetoelastic anisotropy, causing preferred directions of magnetization. For materials with *λ*_s_ > 0, the easy magnetization axis lies along the tensile stress, while for *λ*_s_ < 0 it lies perpendicular to it. If the external magnetic field *H* is generated together with the stress *σ*, the magnetic energy *E* of the system can be expressed as
(1)$$E\left(\varphi \right)=\frac{3}{2}\lambda_{s}\sigma {\left(\sin\varphi \right)}^{2}-\mu_{0}M_{s}H\sin\varphi =\;\frac{3}{2}\lambda_{s}\sigma {\left(\sin\varphi \right)}^{2}-J_{s}H\sin\varphi$$
where is the angle between the magnetization and the direction of the applied stress, *λ*_s_ denotes the saturation magnetostrictive coefficient, and $J_{s}=\mu_{0}M_{s}$ is the saturation magnetic polarization. In the equilibrium state, the energy of the system is minimal, therefore
(2)$$\frac{\partial E}{\;\partial \varphi }=3\lambda_{s}\sigma \sin\varphi \cos\varphi -J_{s}H\cos\varphi =0$$
(3)$$H=\frac{3\lambda_{s}\sigma \sin\varphi }{J_{s}}$$

If the direction of magnetization is perpendicular to *σ* (φ = 90°) then the anisotropy field *H*_a_, connected with induced magnetoelastic anisotropy, can be obtained using the formula: (4)$$H_{a}=\frac{3\lambda_{s}\sigma }{J_{s}}$$

In the Becker–Kersten method the series of hysteresis loops at different stresses, *σ*, are measured. The effective anisotropy field *H*_a_ is defined as a difference between the anisotropy field without applied stress and anisotropy field at an applied stress *H*_a0_ − *H*_aσ_. The saturation magnetostrictive coefficient *λ*_s_ is then obtained by the formula
(5)$$\lambda_{s}=\frac{J_{s}\left(H_{a_{0}}-H_{a_{\sigma }}\right)}{3\sigma }$$

This technique is used above all for long magnetically soft ribbons or transformer steels, whereas *λ*_s_ is obtained as an averaged value from the sample volume placed inside the magnetic field. Its application is typically restricted to samples with negative magnetostriction [7], because in most cases the magnetic field is generated in the same direction as tensile stress and the change in the effective anisotropy field *H*_a_ with and without the applied tensile stress could be zero for positive *λ*_s_ materials.

The bulk saturation magnetostrictive coefficient *λ*_sb_ is positive, so the easy magnetization axis will follow the direction of applied tensile stress. A procedure description for obtaining the surface saturation magnetostriction *λ*_ss_ is presented for the sample of dimension (5 × 8) mm^2^, but it is valid for both the other samples. 

The pressure force of the micrometric screw was gradually adjusted to the values presented in Table 2, hence the surface of the ribbon was exposed to increasing tensile stress *σ* with calculated components *σ*_x_ and *σ*_y_. For every value of pressure force, the surface MOKE hysteresis loop and the group of images with magnetic domains describing ribbon magnetization reversal were measured. Examples of the magnetic domain structure and the hysteresis loops of the sample (5 × 8) mm^2^ for pressure forces 0.1 N and 0.5 N are presented in Figure 9. Moreover, MOKE magnetization curves of the (3 × 8) mm^2^ ribbon are shown in Figure 10. Due to the possibility of changing the plane of light incidence by adjusting an aperture diaphragm of the Kerr microscope, two longitudinal magneto-optical configurations sensitive to the *M*_x_ and *M*_y_ magnetization components were used. The upper panel in Figure 9 and the left subplot of Figure 10 depict the standard configuration sensitive to magnetization component *M*_x_, being parallel with an applied magnetic field and the ribbon axis. For lower levels of stress, the typical rectangular hysteresis loop with a large coercive field little different from the anisotropy field *H*_aσ_ indicates the proximity of the easy magnetization axis. The value of the anisotropy field is precisely determined from the MOKE hysteresis loop, corresponding to the disappearing of the magnetic multidomain structure in the sample. A further increase in the surface tensile stress is responsible for an almost linear decrease in *H*_aσ_, as shown in Figure 11, upper panel. Because of the oblique direction of tensile stress (modelled non-zero components of *σ*_x_ and *σ*_y_), the anisotropy fields of two different applied stresses *σ*_1_ and *σ*_2_ can be expressed as
(6)$$H_{a\sigma 1}=\frac{3\lambda_{\mathrm{ss}}\sigma_{1}\sin\varphi_{1}}{J_{s}},\;H_{a\sigma 2}=\frac{3\lambda_{\mathrm{ss}}\sigma_{2}\sin\varphi_{2}}{J_{s}}$$

By subtracting both expressions, the relation for surface saturation magnetostrictive coefficient *λ*_ss_ is
(7)$$\;\lambda_{\mathrm{ss}}=\frac{J_{s}\left(H_{a_{\sigma 1}}-\;H_{a_{\sigma 2}}\right)}{3\left(\sigma_{1}\sin\varphi_{1}-\sigma_{2}\sin\varphi_{2}\right)}$$

In the magneto-optical configuration sensitive to the *M*_x_ magnetization component, the angle φ between an applied stress *σ* and *M*_x_ is expressed by $\sin\varphi_{1}=\frac{\sigma_{y1}}{\sigma_{1}}$ and $\sin\varphi_{2}=\frac{\sigma_{y2}}{\sigma_{2}}$; the *λ*_ss_ in this configuration is determined by
(8)$$\lambda_{\mathrm{ss}}=\;\frac{H_{a_{\sigma 1}}-\;H_{a_{\sigma 2}}}{\sigma_{y1}-\sigma_{y2}}\frac{J_{s}}{3}=-\frac{\Delta H_{a_{\sigma }}}{\Delta \sigma_{y}}\frac{J_{s}}{3}=-k\frac{J_{s}}{3}$$
where k is the tangent of the line fitting the dependence of *H*_aσ_ on *σ*_y_ (Figure 11, upper panel). The calculated values of *λ*_ss_ for the samples (5 × 8) mm^2^ and (3 × 8) mm^2^ are summarized in Table 4.

The configuration sensitive to the *M*_y_ magnetization component is distinguished from the previous one by applying the magnetic field along the *y* coordinate axis. The plane of incident and the reflected light are adjusted to be parallel with the *y*-axis and only the position of the sample and the applied tensile stress remained unchanged. Examples of the measured surface hysteresis loops and domain patterns are presented in the lower subplot of Figure 9 and the right subplot of Figure 10. Because the easy magnetization axis lies close to the *x*-axis of the ribbon-type sample and the magnetic field is generated almost normally to this direction, the typical near-hard-axis behaviour with substantially higher values of the anisotropy field is observed. It increases nearly linearly with the increasing stress *σ*, as is documented in Figure 11, lower panel. In this case, the angle φ describes the rotation of magnetization from stress *σ* towards the *y* coordinate axis, therefore $\sin\varphi_{1}=\frac{\sigma_{x1}}{\sigma_{1}}$ and $\sin\varphi_{2}=\frac{\sigma_{x2}}{\sigma_{2}}$. As a result, the surface saturation magnetostrictive coefficient *λ*_ss_ depends on the *x*-components of the applied tensile stress according to the relation
(9)$$\lambda_{\mathrm{ss}}=\;\frac{H_{a_{\sigma 1}}\;-\;H_{a_{\sigma 2}}}{\sigma_{x1}-\sigma_{x2}}\frac{J_{s}}{3}=\frac{\Delta H_{a_{\sigma }}}{\Delta \sigma_{x}}\frac{J_{s}}{3}=k\frac{J_{s}}{3}$$
where k is obtained from the linear dependence *H*_aσ_ vs. *σ*_x_ in Figure 11, lower panel. The values of the surface saturation magnetostrictive coefficient *λ*_ss_ for the (5 × 8) mm^2^ and (3 × 8) mm^2^ samples shown in Table 4 are in good agreement with the bulk value *λ*_sb_ (+45.5 ppm) presented in Table 3.

The results of the (10 × 8) mm^2^ sample are illustrated in Figure 12. From Table 2 it is seen that the lowest- and the highest-pressure forces, 1.962 N and 6.867 N, induce the stresses *σ*_x_ = 45.9 MPa, *σ*_y_ = 54.2 MPa and *σ*_x_ = 37.9 MPa, *σ*_y_ = 100.9 MPa, respectively. It is seen that a larger deflection of the sample leads to an increase in the tensile stress in the direction of the *y*-axis and, surprisingly, to a slight decrease in the tensile stress in the direction of the *x*-axis. Therefore, for the estimation of *λ*_ss_, it is more convenient to measure the *M*_x_ magnetization component, it being perpendicular to the increasing *σ*_y_. The observed surface magnetic domains for pressure force 6.867 N, presented in Figure 12, have stripe shapes with walls oriented almost normally to the ribbon axis. At lower magnetic fields, stripe domains of widths up to 100 µm were observed. They vanished at increasing fields and subsequently a fine stripe domain structure was formed closely before saturation. The value of *λ*_ss_, calculated from Equation (8) using the linear dependence *H*_aσ_ vs. *σ*_y_ (not presented here), is comparable with those obtained for other samples, as Table 4 documents. Contrary to this, the configuration which is sensitive to *M*_y_ is not suitable in this case, due to small changes in the tensile stress along the *x*-axis.

## 4. Conclusions

The present study is devoted to the determination of the local surface saturation coefficient of magnetostiction *λ*_ss_ of the thin amorphous ribbon-type sample. The studies are done at the smooth air–ribbon side, because the applied magneto-optical Kerr microscopy is highly sensitive to the surface quality. A special sample holder was developed, produced and successfully used in the present experiment. It has allowed for the inducing of tensile stress in the studied top-surface of the sample due to pressure force acting from the sample’s opposite side. A distribution of local tensile stresses in the sample surface was obtained using the finite element simulation method. A combination of the determined tensile stresses and magnetic characteristics evaluated by the Becker–Kersten method has finally resulted in the determination of the surface magnetostriction. Models of the tensile stress distribution have pointed to an important effect of the sample width. The narrower the sample was, the stronger the uniaxial stress in ribbon axis. The determined values of *λ*_ss_ for 3, 5, and 10 mm wide FeCoB ribbon-type samples have varied between +37 and +60 ppm, in good agreement with the bulk value of +45.5 ppm. The shapes of the local surface hysteresis loops were found to be different by focusing the laser spot into different sample places, documenting dissimilar magnetization reversal and domain wall movements. Nevertheless, the behaviour of the saturation magnetization was practically homogeneous and therefore no marked differences in the *λ*_ss_ values in dissimilar illuminated places were observed.

## Figures and Tables

**Figure 1 materials-13-00257-f001:**
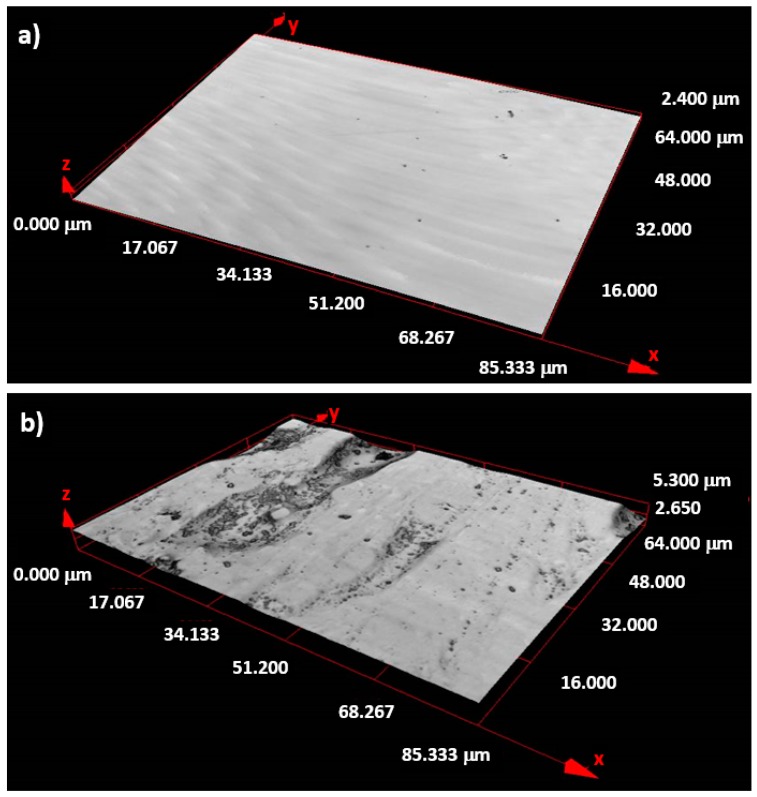
Morphology of the as-prepared FeCoB amorphous ribbon obtained from the (**a**) air and (**b**) wheel surface using the confocal microscope Olympus LEXT 3100.

**Figure 2 materials-13-00257-f002:**
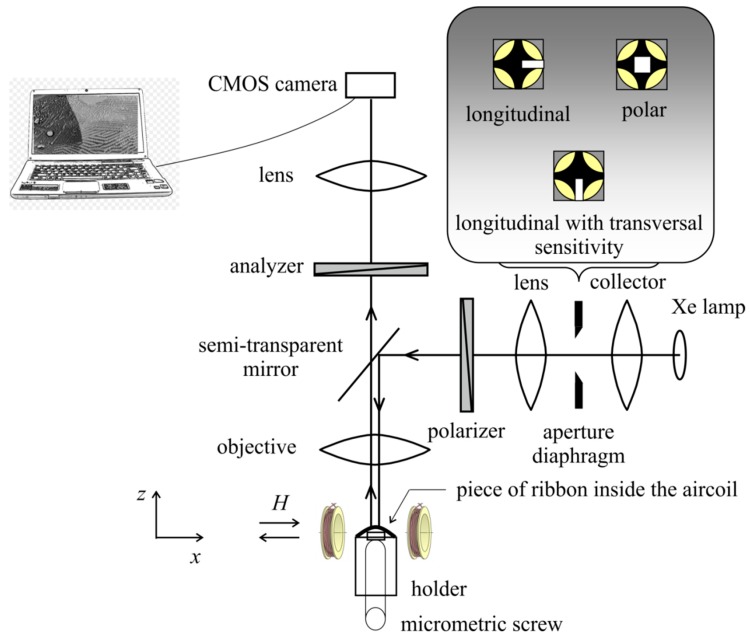
Schema of experiment: magneto-optical Kerr microscope, sample holder, and sample placed into magnetic field produced by couple of coils.

**Figure 3 materials-13-00257-f003:**
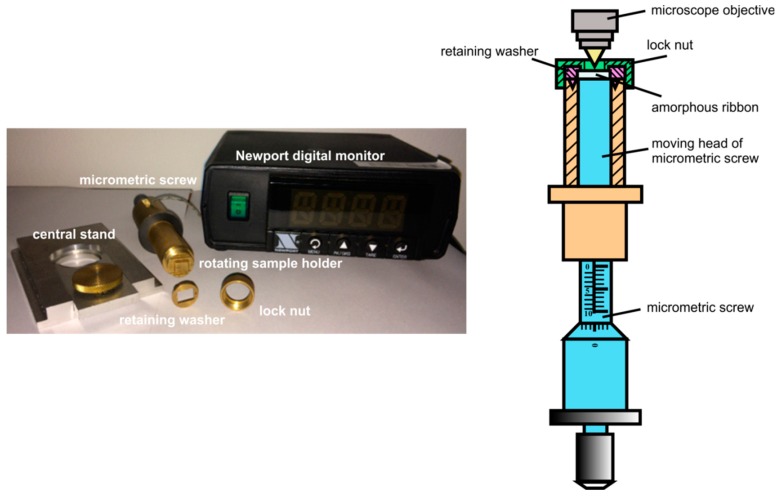
Rotating sample holder for application of tensile stress in the sample surface layers.

**Figure 4 materials-13-00257-f004:**
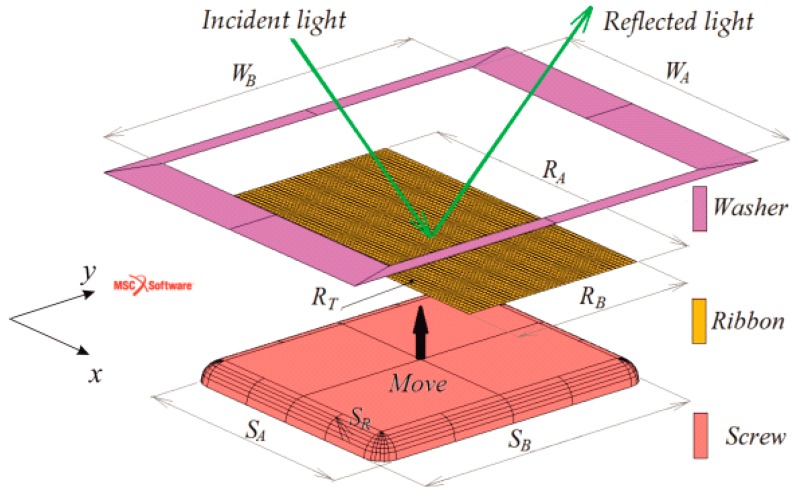
Arrangement of the micrometric screw, ribbon-type sample, and retaining washer for the finite element method (FEM) model. The dimensions are summarized in Table 1.

**Figure 5 materials-13-00257-f005:**
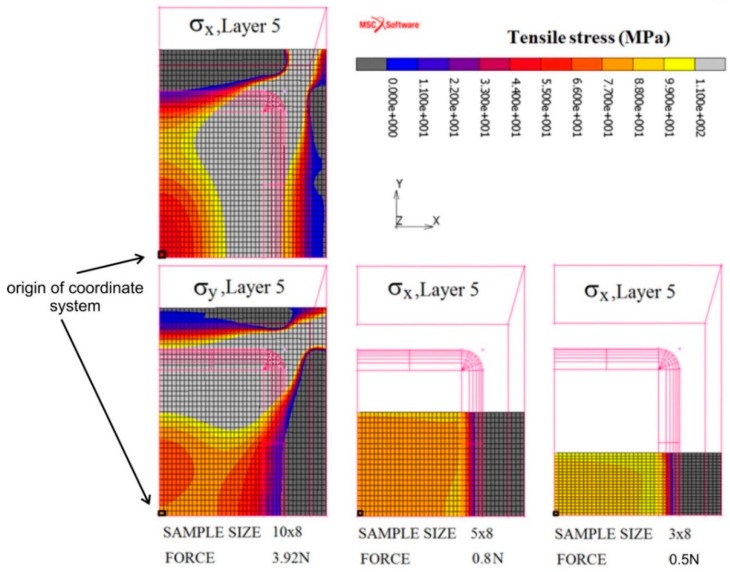
Simulated tensile stresses in x (*σ*_x_) and y (*σ*_y_) directions in the fifth layer of the shell model of the FeCoB samples; *R*_B_ × *R*_A_: 10 × 8 (*σ*_x_,*σ*_y_—left panel), 5 × 8 (*σ*_x_—middle panel), and 3 × 8 (*σ*_x_—right panel). The values at the bottom are pressure forces of the micrometric screw.

**Figure 6 materials-13-00257-f006:**
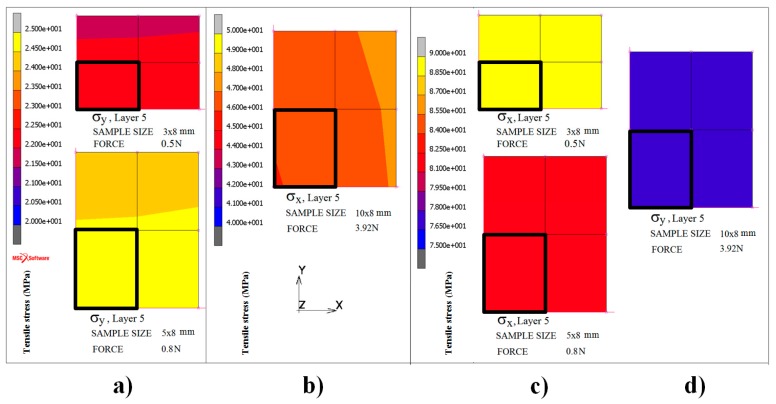
Homogeneous distribution of *σ*_x_ (**b**,**c**) and *σ*_y_ (**a**,**d**) tensile stresses in the area of one element (denoted by black rectangle) on the sample surface.

**Figure 7 materials-13-00257-f007:**
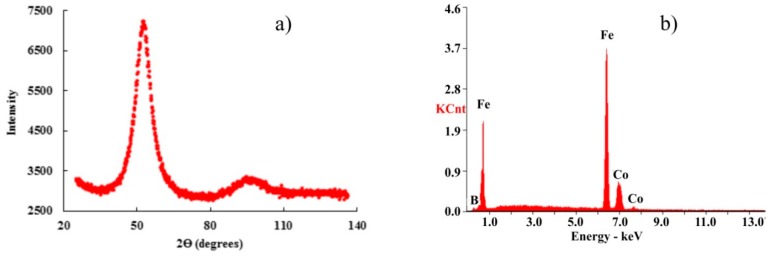
X-ray diffractogram (**a**) and EDX analysis (**b**) both taken from the air-side of the FeCoB ribbon.

**Figure 8 materials-13-00257-f008:**
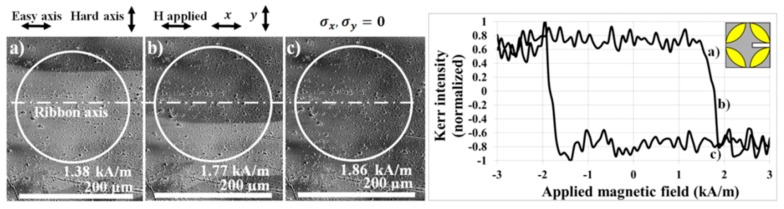
Magnetic domain patterns and corresponding magnetization curve observed using the longitudinal magneto-optical Kerr effect in sample (10 × 8) mm^2^ without applied tensile stress. Subplots (a), (b), and (c) show magnetic domains corresponding to the reversal of the magnetization curve.

**Figure 9 materials-13-00257-f009:**
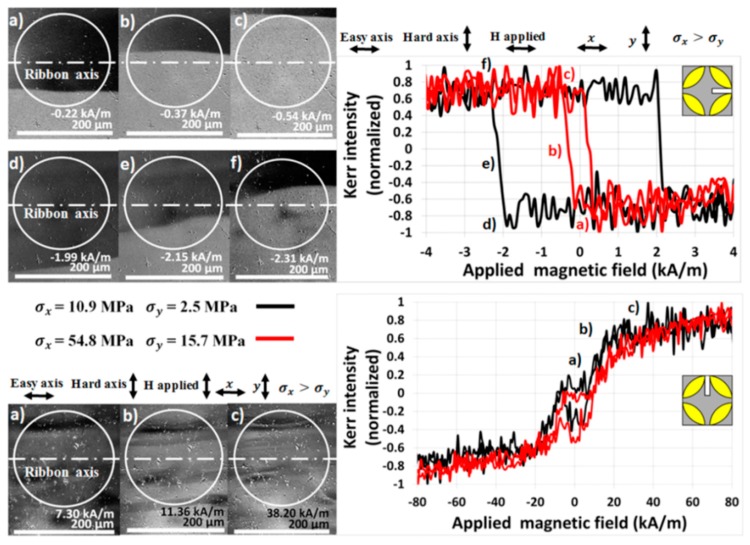
Surface magnetic domains (left) and magnetization curves (right) of the sample (5 × 8) mm^2^ exposed to tensile stress and measured using the longitudinal MOKE configuration related to magnetization *M*_x_ (upper panel) and *M*_y_ (lower panel).

**Figure 10 materials-13-00257-f010:**
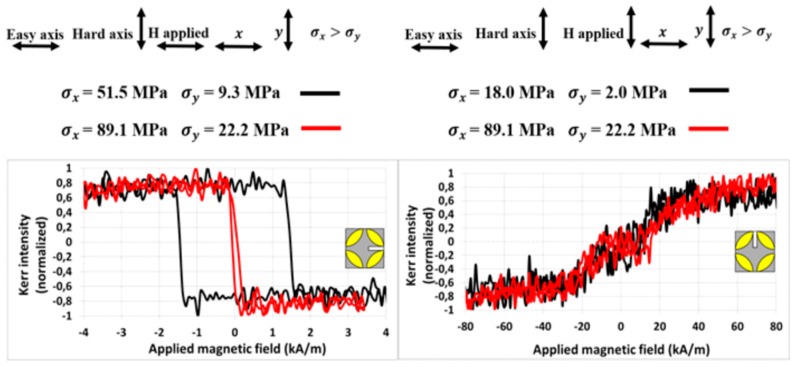
MOKE magnetization curves of the sample (3 × 8) mm^2^ under applied tensile stress. Left and right subplots show sensitivity to the magnetization component *M*_x_ and *M*_y_ and detect the easy and hard magnetization axis.

**Figure 11 materials-13-00257-f011:**
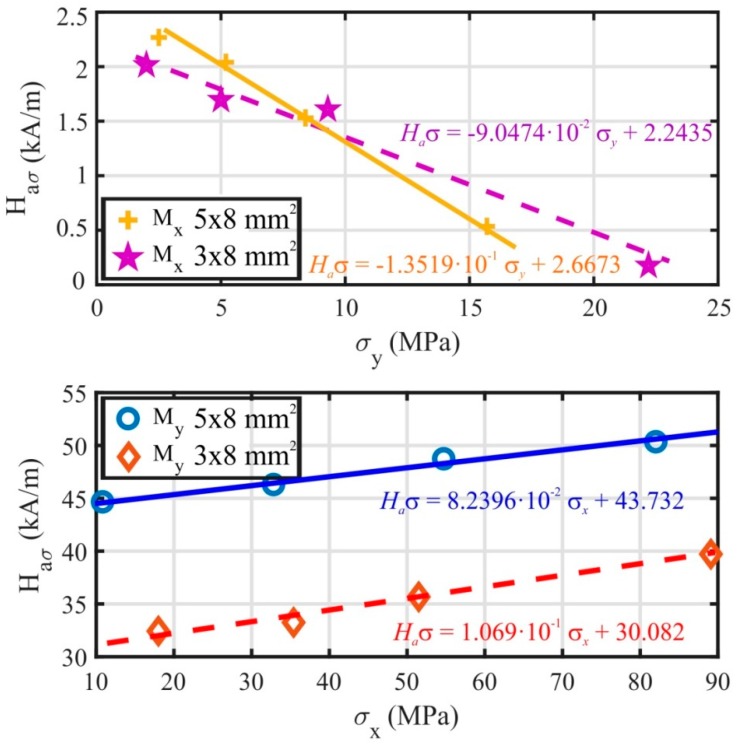
Measured anisotropy magnetic field *H*_aσ_ of FeCoB ribbon depicted as a function of applied stress *σ*_x_ (*σ*_y_) in longitudinal MOKE configurations sensitive to *M*_x_ (upper panel) and to *M*_y_ (lower panel).

**Figure 12 materials-13-00257-f012:**
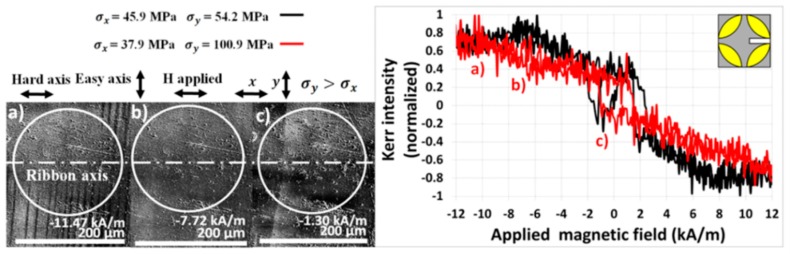
Magnetic domain patterns and magnetization curves (pressure force 6.867 N) of the sample (10 × 8) mm^2^ under applied tensile stress sensitive to the longitudinal *M*_x_ magnetization component.

**Table 1 materials-13-00257-t001:** Dimensions of the FeCoB samples (*R*_A_, *R*_B_, *R*_T_), of the micrometric screw (*S*_A_, *S*_B_, *S*_R_), of the pressure washer (*W*_A_, *W*_B_), and the parameters for modelling of tensile stress in the surface layer.

Exp	*R*_A_ [mm]	*R*_B_ [mm]	*R*_T_ [µm]	*S*_A_ [mm]	*S*_B_ [mm]	*S*_R_ [mm]	*W*_A_ [mm]	*W*_B_ [mm]	Pressure Force of the Washer [N]	Element [mm]	Number of Elements	Number of Nodes
1	8	10	34	6	8	0.5	7.2	9.2	2 × 40 × 4 × 0.0001	0.1 × 0.125	1600	1682
2	8	5							20 × 4 × 0.028	0.1 × 0.128	800	862
3	8	3							20 × 4 × 0.017	0.1 × 0.075	800	862

**Table 2 materials-13-00257-t002:** Measured values of pressure force of micrometric screw used in the FEM model (Column 1) and parameters obtained from the applied model (Columns 2–5).

Pressure Force of Micrometer Screw (N)	Deflection of Sample (mm)	Modelled Micrometric Screw Shift (mm)	Tensile Stress in the	
			*x*-axis *σ_x_* (MPa)	*y*-axis *σ_y_* (MPa)
**Sample dimension (*R*_B_ × *R*_A_) 10 × 8 mm—experiment 1**				
1.962	0.152	0.088	45.9	54.2
2.943	0.183	0.110	47.2	67.6
3.924	0.208	0.128	45.8	78.1
4.905	0.227	0.142	43.8	86.0
5.886	0.244	0.156	40.8	94.1
6.867	0.256	0.168	37.9	100.9
**Sample dimension (*R*_B_ × *R*_A_) 5 × 8 mm—experiment 2**				
0.100	0.020	0.010	10.9	2.5
0.300	0.068	0.030	32.9	8.4
0.500	0.112	0.049	54.8	15.7
0.800	0.174	0.078	82.1	24.6
**Sample dimension (*R*_B_ × *R*_A_) 3 × 8 mm—experiment 3**				
0.100	0.038	0.017	18.0	2.0
0.200	0.077	0.034	35.4	5.0
0.300	0.112	0.049	51.5	9.3
0.500	0.205	0.094	89.1	22.2

**Table 3 materials-13-00257-t003:** Bulk magnetic parameters of the FeCoB ribbon: *J*_s_—saturation magnetic polarization; *H*_c_—coercive field; *λ*_sb_—bulk saturation magnetostrictive coefficient.

*J*_s_ (T)	*H*_c_ (A/m)	*λ*_sb_ (ppm)
1.35	15.12	+45.5 [18]

**Table 4 materials-13-00257-t004:** Local surface saturation magnetostrictive coefficient *λ*_ss_ of FeCoB ribbon calculated in longitudinal MOKE configurations sensitive to magnetization components *M*_x_ or *M*_y_.

Sample Dimension *R*_B_ × *R*_A_ (mm^2^)	Magnetization Component	*λ*_ss_ (ppm)
5 × 8	*M* _x_	+60.84
	*M* _y_	+37.08
3 × 8	*M* _x_	+49.88
	*M* _y_	+40.71
10 × 8	*M* _x_	+49.53

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
