# Peer review of "Surface Magnetostriction of FeCoB Amorphous Ribbons Analyzed Using Magneto-Optical Kerr Microscopy"

_materials, 2020, doi:10.3390/ma13020257_

Round 1

Reviewer 1 Report

In experimental details, I felt authors should improve Figures 8 and 10 based on Table 2 to make it easier for readers to understand. For example, authors would also need to show data where the magnetic field was applied in the direction of the easy axis in the magnetic domain pattern and magnetization curve in Figure 10. In addition, I am also interested in the behavior of the magnetization curves of Sample (3x8) mm2.  

In conclusions, I want authors to further explain the fact that the behavior of the saturation magnetization is virtually uniform, despite the different shapes of the local surface hysteresis loops.

Author Response

We thank to the Reviewers for their comments leading to improvement of the manuscript. It was modified according to the reviewer’s suggestions. All changes are denoted by yellow colour in the revised version of the manuscript and answers on the reviewer’s comments are summarized in the following text:

Reviewer 1:

Comment

In experimental details, I felt authors should improve Figures 8 and 10 based on Table 2 to make it easier for readers to understand. For example, authors would also need to show data where the magnetic field was applied in the direction of the easy axis in the magnetic domain pattern and magnetization curve in Figure 10. In addition, I am also interested in the behavior of the magnetization curves of Sample (3x8) mm2.

Reply

According to the reviewer suggestion we have added Figure 10 showing magnetization curves of the sample (3x8) mm2 with its caption and implemented it into the text. We believe that due to this figure text will be more understandable for the readers.

The conclusions of our manuscript show that surface saturation magnetostriction coefficient depends on the tangent of the line fitting the dependence of Hon σ. If the measuring setup is sensitive to the My magnetization component, saturation magnetostriction coefficient is influenced by the σx changes, while for the Mx magnetization component it depends on the σy changes. Modelled tensile stresses of the sample (10x8) mm2 show the biggest changes in the y-axis, therefore the measuring setup was adjusted for the Mx magnetization component detection and big changes of the anisotropy field were measured. Conversely, the changes of tensile stress in the x-axis are low (Dσx » 10 MPa) in comparison to the y-axis and also to the other samples in both axis. As a result calculated magnetostriction coefficient in saturation would be affected by the big error due to small changes of anisotropy field by the applied tensile stress (curves are also influenced by the noise from the surface roughness). Therefore, experiment sensitive to the My magnetization component in the case of the sample (10x8) mm2 was not done.  

Comment

In conclusions, I want authors to further explain the fact that the behavior of the saturation magnetization is virtually uniform, despite the different shapes of the local surface hysteresis loops.

Reply

In the case of amorphous and/or nanocrystalline ribbons prepared by planar flow technique we observe inhomogeneous surface magnetic properties of these materials using MOKE. It means that if the laser spot is focused into the different sample places, local domain structure is different in each place due to local magnetic anisotropy (tensile and compressive stresses from preparation process) and existing surface defects. As a result magnetization reversal observed at lower magnetic fields on the magnetization curve is different for each illuminated place. This could lead to the question if surface saturation magnetostriction coefficient could be also different in every illuminated place. Our conclusions confirm that although the values of anisotropy field can be little bit different, tangent of dependence of Hon σ determining size of the saturation magnetostriction coefficient is almost constant.        

Reviewer 2 Report

referee report
materials 675318
Surface magnetostriction of FeCoB amorphous ribbons analyzed using the magneto-optical Kerr
microscopy
Kamila Hrabovská, Ondřej Životský, Jaroslav Rojíček, Martin Fusek, Vratislav Mareš and
Yvonna Jirásková

This manuscript reports on surface magnetostriction of FeCoB amorphous ribbons, observed by
employing the magneto-optic Kerr effect. The topic is well fitting to the journal Materials.

The manuscript is mostly well written and well organized. The English is satisfactory. The reference list is fairly complete,
and the data presented support the conclusions drawn.

There are, however, some points which need clarification and improvement:

Experimental details: No model number of the Zeiss microscope is given, no wavelength of the
polarized light. As the samples were analyzed in their as-grown state, no ZnS layer was applied.
Again, this should be commented upon.

The samples were obviously used in the as-grown state. The authors provide surface data, Ra and Rz, but
no image of the sample. Why not? This would be much more informative for the readers, e.g., in form of
a color image with the colors giving the z-scale.
Furthermore, the authors should explain why they wanted to investigate the samples in their as-grown
state and why no attempt to polish the sample surface was made.

The section on the sample holder for the investigations is very nice, so the lack of a complete
image of the domain structure of the sample is even more ununderstandable.

Overall, the manuscript can be published in Materials, if the points mentioned above are treated well.

Author Response

We thank to the Reviewers for their comments leading to improvement of the manuscript. It was modified according to the reviewer’s suggestions. All changes are denoted by yellow colour in the revised version of the manuscript and answers on the reviewer’s comments are summarized in the following text:

Reviewer 2:

Comment

No model number of the Zeiss microscope is given, no wavelength of the polarized light. As the samples were analyzed in their as-grown state, no ZnS layer was applied. Again, this should be commented upon.

Reply

In section 2.2 (page 3 of the manuscript) we have added model of the used Zeiss microscope – AxioImager M1 – and also information about source of a light (white light from Xe lamp containing mix of all visible wavelengths). Ribbons were prepared by planar flow casting technique in the as-prepared state, so we do not need any ZnS layer for their preparation or observation of magnetic domains using the magneto-optical Kerr microscopy from the air surface. Therefore, there is no mention of the ZnS layer in the manuscript.

Comment

The samples were obviously used in the as-grown state. The authors provide surface data, Ra and Rz, but no image of the sample. Why not? This would be much more informative for the readers, e.g., in form of a color image with the colors giving the z-scale.

Reply

We have added Figure 1 showing morphology of the investigated ribbon from its both sides. This is image available from the confocal microscope Olympus LEXT 3100.

Comment

Furthermore, the authors should explain why they wanted to investigate the samples in their as-grown state and why no attempt to polish the sample surface was made.

Reply

Aim of the paper is mainly introduction of new technique suitable and sensitive enough for estimation of surface saturation magnetostriction coefficient. Generally, this technique can be used for any magnetic ribbon prepared by planar flow casting technique. We used FeCoB because of its low positive magnetostriction in as-prepared state, so evaluation of the magnetostriction using the Becker-Kersten method is then more complicated.

Magnetic domain patterns observed in different places on the ribbon surface are not the same and as a result the magneto-optical hysteresis loops have different shapes. But polishing of the sample would destroy these original domain patterns and it would bring its own anisotropy on the ribbon surface. This is the main reason, why polishing was not applied.

Comment

The section on the sample holder for the investigations is very nice, so the lack of a complete image of the domain structure of the sample is even more ununderstandable.

Reply

Section 3.1 is devoted to the sample holder description, while magnetic domain patterns of the ribbon are shown in section 3.2 – in Fig. 8 without applied tensile stress, while in Figures 9 [samples (5x8 and 3x8) mm2] and 11 [sample (10x8) mm2] with applied tensile stress.